# Comparative Transcriptome Analysis of Pine Trees Treated with Resistance-Inducing Substances against the Nematode *Bursaphelenchus xylophilus*

**DOI:** 10.3390/genes11091000

**Published:** 2020-08-26

**Authors:** Jungwook Park, Hee Won Jeon, Hyejung Jung, Hyun-Hee Lee, Junheon Kim, Ae Ran Park, Namgyu Kim, Gil Han, Jin-Cheol Kim, Young-Su Seo

**Affiliations:** 1Department of Integrated Biological Science, Pusan National University, Busan 46241, Korea; jjuwoogi@pusan.ac.kr (J.P.); jhj4059@pusan.ac.kr (H.J.); ehyuna92@pusan.ac.kr (H.-H.L.); titanic622@pusan.ac.kr (N.K.); croone@pusan.ac.kr (G.H.); 2Environmental Microbiology Research Team, Nakdonggang National Institute of Biological Resources (NNIBR), Sangju 37242, Korea; 3Department of Agricultural Chemistry, Institute of Environmentally Friendly Agriculture, College of Agriculture and Life Sciences, Chonnam National University, Gwangju 61186, Korea; jeon-hw@naver.com (H.W.J.); arpark9@naver.com (A.R.P.); 4Forest Insect Pests and Diseases Division, National Institute of Forest Science, Seoul 02455, Korea; junheonkim@korea.kr

**Keywords:** comparative transcriptome, pine wilt disease, *Bursaphelenchus xylophilus*, *Pinus densiflora*, systemic acquired resistance (SAR), acibenzolar-*S*-methyl (ASM), methyl salicylic acid (MeSA)

## Abstract

The pinewood nematode (PWN) *Bursaphelenchus xylophilus* causes pine wilt disease, which results in substantial economic and environmental losses across pine forests worldwide. Although systemic acquired resistance (SAR) is effective in controlling PWN, the detailed mechanisms underlying the resistance to PWN are unclear. Here, we treated pine samples with two SAR elicitors, acibenzolar-*S*-methyl (ASM) and methyl salicylic acid (MeSA) and constructed an in vivo transcriptome of PWN-infected pines under SAR conditions. A total of 252 million clean reads were obtained and mapped onto the reference genome. Compared with untreated pines, 1091 and 1139 genes were differentially upregulated following the ASM and MeSA treatments, respectively. Among these, 650 genes showed co-expression patterns in response to both SAR elicitors. Analysis of these patterns indicated a functional linkage among photorespiration, peroxisome, and glycine metabolism, which may play a protective role against PWN infection-induced oxidative stress. Further, the biosynthesis of flavonoids, known to directly control parasitic nematodes, was commonly upregulated under SAR conditions. The ASM- and MeSA-specific expression patterns revealed functional branches for myricetin and quercetin production in flavonol biosynthesis. This study will enhance the understanding of the dynamic interactions between pine hosts and PWN under SAR conditions.

## 1. Introduction

Pine trees from the genus *Pinus* are widely distributed in a variety of environments, ranging from areas at sea level to mountainous areas and from some of the coldest to the hottest environments on earth [1,2,3]. Pine forests are among the most valuable resources, with respect to timber, fuel, and various wood products [4,5,6]; native pine forests also provide important habitats or source of food for numerous species of birds and mammals in natural ecosystems. However, in recent decades, pine trees have suffered from epidemic attacks of pine wilt disease (PWD), and their populations have reduced markedly since the 1960s [7,8]. PWD, caused by the pinewood nematode (PWN) *Bursaphelenchus xylophilus*, is an extremely destructive disease that affects pine forests and results in substantial economic and environmental losses on a worldwide scale. Subsequent to its initial detection in Japan at the beginning of the 20th century [9], PWD spread rapidly throughout a number of East Asian countries, notably, China, Korea, and Taiwan, and thereafter extended to Europe (Portugal and Spain) [10,11,12,13]. In Japan, some 700,000 m^3^ of pine forests are infected by PWD each year [14], whereas in the 10 years between 2003 and 2013, China lost USD 20 billion as a consequence of the mass dieback of 1 million ha of pine forests in 16 provinces [15].

*B. xylophilus* has the unique capacity to feed on both living trees and fungi [16,17]. Moreover, it is known to be vectored by pine sawyers of the genus *Monochamus* and is transmitted from deceased to healthy pine trees during the summer period of oviposition [18,19]. The PWN feeds on epithelial cells in the resin canals of host trees and subsequently migrates through surrounding tissues to kill parenchymal cells [20,21,22]. Populations of infesting PWN undergo rapid expansion, and in the process cause extensive destruction of xylem tissues, thereby interrupting water and resin flow in host trees. By autumn, the trees have usually succumbed to the detrimental effects of the infection [18,19]. The reiteration of infection and the pathogenic cycle ensures that the PWN is readily disseminated throughout pine forests.

Owing to the serious threat posed by PWN, considerable effort has been expended in attempts to control the disease. Control methods are based primarily on aerial and ground applications of conventional insecticides (fenitrothion and thiacloprid) to kill the pine sawyer vectors, or involve the injection of trees with nematicides, such as levamisole hydrochloride, morantel tartrate, and emamectin benzoate [23,24,25]. Recently, however, there has been growing public concerns regarding the use of pesticides and nematicides, on account of their detrimental effects on the environment and human health, as well as undesirable effects on non-target organisms [23,26]. In this regard, the establishment of systemic acquired resistance (SAR) in pine trees may provide a viable alternative that can overcome the problems associated with the currently used control agents. SAR is a highly enhanced immune-type plant resistance that confers long-term broad-spectrum resistance against a diverse range of invaders [27,28,29,30]. Prophylactic induction of SAR prior to contact with nematodes enables pine trees to mount a more effective resistance against PWD [8,31]. SAR can be induced not only by exposure of plants to abiotic and biotic stresses but also by using synthetic resistance-inducing substances (RISs) [32]. Among the commonly employed RISs, salicylic acid (SA) is an endogenous signalling compound that regulates the induction of SAR, as well as plant growth and development [33,34,35]. Although the efficacy of SAR has been well validated, numerous details remain unclear, notably, the identity of the genetic elements activated by RISs, and specifically in the case of PWD, how to effectively exploit SAR to protect pine trees from PWN infestation.

Gaining an understanding of the mechanisms underlying resistance to PWD is a complex issue and demands integrated management, as studies that focus exclusively on isolated elements of the host–pathogen system (i.e., host plant, SAR, or the causal nematode) fail to fully reflect the dynamic interactions among these key elements. Ideally, in vivo studies on PWN-infected pine trees under conditions of induced SAR are the optimal approach for addressing this problem. In this regard, recent advances in high-throughput sequencing have enabled unbiased and more accurate quantification of transcripts at a high resolution to determine genome-wide expression profiles in organisms under specified conditions [36]. In vivo transcriptome analyses have made it possible to trace the cascades of biological systems that are triggered in response to invaders, and to identify key defence-related functions in plant–nematode interactions [37,38]. Furthermore, comparison between different in vivo transcriptomes has enabled us to assess the fundamental principles of plant resistance against pathogens, as well as the specific effects of the gene expression patterns uniquely induced under each condition [39,40,41].

In this study, we examined the effects of two SA-derived SAR elicitors, acibenzolar-*S*-methyl (ASM) and methyl salicylic acid (MeSA), which effectively inhibited disease symptoms in PWN-infected pine trees. After treatment with these elicitors, the in vivo differentially expressed genes (DEGs) were characterised in *B. xylophilus*-infected pine trees. Compared to control samples that were not treated with elicitors, DEGs reflected both the stressful conditions under highly diseased symptoms and the defence response against nematode infection. A comparative analysis of these in vivo transcriptomes revealed that co-expression patterns among upregulated DEGs constitute major biological systems in pine trees. Furthermore, by performing network analysis based on protein–protein interactions, we validated key functions, including photorespiration and flavonoid biosynthesis, which may play a critical role in the inhibition of, and protection against, PWN. In addition, we conducted functional enrichment analysis of unique expression patterns to elucidate the ASM- and MeSA-specific reactions and outputs that comprise the defence mechanisms of pine trees. Collectively, the findings of this study provide new insights into the positive roles of RISs during PWD development and contribute to enhancing general understanding of the interactions between host pine trees and PWN.

## 2. Materials and Methods

### 2.1. PWN Source

The causal agent of PWD, the PWN *B. xylophilus*, was obtained from the National Institute of Forest Science (NIFoS: Seoul, Korea). *B. xylophilus* was isolated from the infected tissues of red pine (*Pinus densiflora*) and is highly virulent to genus *Pinus* trees [42,43]. For all experiments, PWN was cultured on the hyphae of the fungus *Botrytis cinerea* grown on potato dextrose agar medium (PDA) at 25 °C for 1 week. Cultured PWN was separated from the PDA medium using the Baermann funnel technique [44]. A suspension of PWN in distilled water was adjusted to a concentration of approximately 20,000 nematodes/mL.

### 2.2. Preparation of Chemical Formulations

ASM (>99.4% purity) was kindly supplied by Syngenta Korea (Seoul, Korea). A suspension concentrate-type formulation of ASM was prepared by mixing 5.03% (*w*/*w*) ASM, 2% (*w*/*w*) CR-KSP40M (the mixture of ethoxylated tristyrylphenol phosphate, potassium salt, and propylene glycol) as a wetting and dispersing agent, 10% (*w*/*w*) propylene glycol as an antifreeze agent, 0.1% (*w*/*w*) xanthan gum as a viscosity modifier, 0.1% (*w*/*w*) 1,2-Benzisothiazolin-3-one as a preservative, 0.1% (*w*/*w*) polydimethylsiloxane as a defoamer, and 82.67% (*w*/*w*) water. MeSA (>99.0% purity) was purchased from Tokyo Chemical Industry Co., Ltd. (Tokyo, Japan), an emulsifiable concentrate-type formulation which was prepared by mixing 20% (*w*/*w*) MeSA, 20% (*w*/*w*) propylene glycol mono-methyl ether as a solubilizer, 20% (*w*/*w*) CR-MOC25 (the mixture of ethoxylated castor oil, calcium dodecylbenzenesulfonate, and tristyrylphenol ethoxylates) as an emulsifier, and 40% (*w*/*w*) methylated soybean oil as a diluent.

### 2.3. In Vivo Pine Seedling Assay

For use in in vivo assays, *P. densiflora* seedlings (3 years old) were purchased from Daelim seedling farm (Okcheon, Korea), and maintained under a relative humidity of 70%. ASM and MeSA preparations were dissolved in water at an active concentration of 1600 μg/mL, of which 5 mL was subsequently sprayed onto the pine seedlings, with the treatment being conducted twice at 1-week intervals. Distilled water containing 250 μg/mL Tween 20 was used as an untreated negative control. One week after treatment, the pine seedlings were inoculated with PWN. Briefly, after making a small slit with a surface-sterilised knife, a small piece of absorbent cotton was embedded into the slit, and 100 μL of PWN suspension was pipetted onto the absorbent cotton. The inoculation sites were then covered with the parafilm (Heathrow Scientific, Vernon Hills, IL, USA) to prevent desiccation. Each treatment was replicated four times and each experiment was conducted twice. At 30 days after PWN inoculation, PWD was evaluated with reference to a numerical scale from 0 to 5, where 0 = healthy seedlings with normal green needles; 1 = 1–20% discoloured (brown) needles; 2 = 21–40% brown needles; 3 = 41–60% brown needles; 4 = 61–80% brown needles with bending of the terminal shoots of seedlings; 5 = 81–100% brown needles and wilting of the entire seedling. Disease severity was determined as follows: disease degree = ∑ (number of samples per score × score)/total number of needles. Pine seedlings that were not inoculated with PWN served as the positive controls.

### 2.4. RNA Isolation

*Pinus* tree RNAs for in vivo transcriptome analysis were extracted from samples obtained from plants used in the pine seedling assay. Pine samples were surface sterilised with 70% ethanol and then rinsed with distilled water. Plant RNAs were isolated using a modified hexadecyltrimethylammonium bromide (CTAB) method [45]. One gram of needles was ground in liquid nitrogen and mixed with 15 mL of extraction buffer containing 100 mM Tris-HCl (pH 8), 2% CTAB, 30 mM ethylenediaminetetraacetic acid, 2 M NaCl, 0.05% spermidine, 2% polyvinylpolypyrrolidinone, 2% 2-mercaptoethanol, and 1.5 mg/mL proteinase K. The suspension was incubated at 42 °C for 90 min and then extracted with 15 mL chloroform-isoamyl alcohol (24:1). Total RNAs were precipitated with 10 M LiCl. After washing with 70% ethanol, the RNA pellets were dissolved in diethylpyrocarbonate-treated water, and RNA quality was evaluated using gel electrophoresis and a NanoDrop2000 spectrophotometer (Thermo Scientific, Barrington, IL, USA).

### 2.5. RNA Sequencing

An Agilent 2100 Bioanalyzer (Agilent Technologies, Santa Clara, CA, USA) was used to assess RNA integrity. Total RNAs were converted to cDNA libraries using a TruSeq RNA Sample Preparation Kit (Illumina, San Diego, CA, USA). Briefly, 1 μg of total RNAs was enriched for mRNA using oligo-dT coated magnetic beads, fragmented, and reverse transcribed into cDNAs. The cDNA libraries were sequenced using an HiSeq 2000 sequencing platform (Illumina), according to the manufacturer’s instructions, by Macrogen (Seoul, Korea). RNA-seq libraries comprising sequences with 101-bp read lengths were constructed using the paired-end strategy. The quality of raw reads was evaluated using the FastQC tool [46]. All raw sequencing data presented in this study have been deposited in the National Center for Biotechnology Information Gene Expression Omnibus database under accession no. GSE154134.

### 2.6. Genome Characterization

The reference genome sequence for loblolly pines from the genus *Pinus* was downloaded from the PineRefSeq database [47], and the gene information used for transcriptome analyses was derived via v2.01 annotation [48]. Genes, exons, and coding sequences were defined in the downloaded GTF file. Duplicate gene information predicted using the annotation tools Augustus, GenomeThreader, and Gmap was filtered using an in-house-developed Python script.

### 2.7. Transcriptome Analysis

Raw reads in the FASTQ format were filtered using the FASTX-Toolkit [49] to remove sequences with low-quality scores (50% of the sequence with a Phred score ≥ 28). An in-house Python script was used to synchronise the forward and reverse directions of the filtered reads. Pre-processed reads were aligned to the reference genome sequence using the Burrows-Wheeler Aligner (BWA) based on the maximal exact matches algorithm [50]. SAMtools was used to convert SAM files containing mapping results to binary format BAM files, and subsequently we sorted these according to genomic positions [51]. Mapped reads per coding sequence in *Pinus* trees were counted using the FeatureCounts module in the Subread package [52]. To quantify gene transcription levels, we determined relative transcript abundance using the reads per kilobase per million mapped reads (RPKM) method [53].

### 2.8. DEG Analysis

Analyses of the ASM- and MeSA-specific DEGs were performed using the DEGseq package in the R statistical environment [54]. In this statistical package, MARS with a random sampling model is widely used to detect and visualise the intensity-dependent ratio of transcriptome data [55]. As input data, we used the RPKM values of each gene, and thereafter examined the effects of the two RISs on genome-wide gene expressions compared with the untreated controls. For multiple testing corrections, *p*-values calculated for each gene were adjusted to false discovery rate (FDR) values using the Benjamini–Hochberg procedure [56]. The criteria set to indicate a significant difference in expression were as follows: |log_2_-fold change| ≥ 1 and FDR < 0.05. The functional characterisation of all DEGs was performed using the BlastKOALA tool, selecting the eukaryotes taxonomy group [57]. On the basis of the identified Kyoto Encyclopedia of Genes and Genomes (KEGG) ortholog groups of each DEG, we constructed BRITE functional hierarchies and KEGG pathways.

### 2.9. Comparative Transcriptome and Functional Enrichment

For a comparative study of the transcriptomes obtained following the ASM and MeSA treatments, we implemented the VennDiagram package in the R environment, which can depict the co-expression and unique groups of upregulated DEGs.

On the basis of the KEGG pathway and gene ontology (GO) terms, we performed functional enrichment analyses to identify the significantly activated systems in the co-expression and unique patterns. KEGG and GO information for all pine genes were obtained from the annotation file of the PineRefSeq database. We assessed the scale of upregulated DEGs for each system, and the hypergeometric distribution was used to identify significantly enriched systems using the phyper function in R [58]. The parameters used in this analysis were as follows: *N*, the number of genes annotated to KEGG and GO information; *n*, the number of DEGs of *N*; *M*, the number of genes annotated to a specific system; *m*, the number of DEGs of *M*. An FDR value ≤ 0.05 was considered to be indicative of a significant difference. To enhance system stability, results with less than five genes were filtered out. Network structures of functional systems of interest were also constructed using the Cytoscape tool [59].

### 2.10. Protein–Protein Interaction Network Analysis

The upregulated DEGs associated with the co-expression patterns derived from comparative transcriptome analysis were used as input data for the Search Tool for the Retrieval of Interacting Genes/Proteins (STRING) database [60]. STRING v11 contains 24,584,628 proteins from 5090 organisms and facilitates assessments of functional interactions, including all types of direct and indirect associations. For the purpose of this analysis, we selected Tracheophyta, as the higher taxonomic level, as the background source. The minimum required score for each interaction was customised at a confidence criterion of 0.50.

### 2.11. Quantitative Real-Time PCR (qPCR) Analysis

Total RNAs were extracted from the pine samples under the same conditions as the in vivo transcriptome analysis. First-strand cDNA was synthesised by using SuperiorScript™ III cDNA Synthesis Kit (Enzynomics, Daejeon, Korea) with oligo-dT primers in accordance with the manufacturer’s protocol. Several genes belonging to major systems in co-expression patterns were subjected to qPCR with specific primers (Appendix A). qPCR analysis was performed with TOPreal™ qPCR 2X PreMix (SYBR Green with low ROX) (Enzynomics) on a Rotor-gene Q (Qiagen, Valencia, CA, USA). Relative expression levels were calculated using the comparative 2^−ΔΔCT^ method with elongation factor (PITA_47202) as the internal control. The qPCR results are shown as the means ± SD of three independent biological replicates.

## 3. Results

### 3.1. Inhibitory Effects of the ASM and MeSA Treatments against PWN Infection

Inhibitory activity under conditions of SAR induction was assessed during the development of PWD. Following the treatment of 3-year-old pine seedlings with either ASM or MeSA, the severity of disease (as ascertained by needle discolouration) caused by *B. xylophilus* was evaluated (Figure 1). On day 30 after treatment, untreated pine seedlings showed an infection rate of 100%, with a disease severity score of 5. In contrast, seedlings treated with RISs showed substantially fewer PWD-related symptoms, with most needles remaining healthy with the absence of brown discolouration, and overall disease severity being scored as 0.75 (an average infection rate of 15%). All seedlings that received no PWN inoculation remained healthy. These results accordingly indicated that ASM- and MeSA-induced SAR were effective in activating defence mechanisms against PWD.

### 3.2. Overview of the In Vivo Transcriptomes in PWN-Infected Pine Trees

After identifying the positive effects of ASM and MeSA with regard to resistance against *B. xylophilus* infection (Figure 1), an in vivo transcriptome analysis of pine trees was performed using RNA-seq technology. SAR was induced in pine seedlings via treatment with ASM and MeSA, and thereafter PWN-infected seedlings were harvested to extract in vivo total RNAs. Pines treated with distilled water containing Tween 20 (untreated sample) were used as negative controls. From a total of nine RNA-seq libraries with three replicates, approximately 294 million paired-end reads were generated, yielding an average of 32 million reads per library (Table 1). Most reads were of good quality with an average Phred score of 40 (Appendix A). After quality filtering, a total of 288 million reads (98.0%) remained (Appendix A). A reference genome sequence for the genus *Pinus* was previously reported by Zimin et al. [48], the 22,103,635,615-bp assembly of which has a GC content of 34.8% and 47,602 computationally predicted genes, among which, 45,528 are non-redundant genes (Appendix A). The filtered reads were mapped onto the reference genome sequences using BWA tool [50], with an average of 89.2% (ranging from 23,843,785 to 30,506,098) of all reads being successfully mapped (Appendix A).

### 3.3. Genes Induced by PWN Infection under SAR Induction

During the course of PWD development in pine trees, changes in genome-wide gene expressions in response to treatment with RISs were investigated. The 45,528 non-redundant genes were analysed by applying |log_2_-fold change (RIS sample/untreated sample)| ≥ 1 and FDR < 0.05 as cut-off criteria (Figure 2a,b). In scatter plots and M (log ratio) against A (mean average) scales (MA-plots), each point represents a single pine gene. We compared the expression level of each gene in the untreated sample with that in ASM and MeSA treatment samples. In the MA-plots, red points indicated candidates that have an FDR value < 0.05. All DEGs in each of the two RIS treatments are listed in Appendix A. In response to ASM treatment, 1915 genes were differentially expressed during PWN infection (Figure 2c and Appendix A), among which, 1091 (57.0%) were upregulated and 824 (43.0%) were downregulated, whereas in seedlings treated with MeSA, there were 1139 (60.6%) upregulated and 742 (39.4%) downregulated DEGs (Figure 2c and Appendix A). Since the major physiological activity would have been inactivated in the stressful conditions of the untreated samples, it seems that a relatively large number of genes were distributed with upregulation changes.

It was confirmed that the DEGs induced by the two RISs are distributed in the same major functional hierarchies, although the orders differed. In the donut charts depicted in Figure 2c, the outer rings show the top five functional categories and their respective proportions. DEGs encoding carbohydrate metabolism comprised the largest functional category in both ASM (123 genes, 16.0%) and MeSA (131 genes, 17.6%) treatments. Other major categories included are as follows: biosynthesis of other secondary metabolites (85 and 66 genes in the ASM and MeSA treatments, respectively); lipid metabolism (67 and 53 genes); protein families: genetic information processing (65 and 61 genes); protein families: signalling and cellular processes (58 and 70 genes). These findings revealed that the genetic elements in the pine genome induced by the two RISs for defence against PWN tend to be quantitatively and functionally similar.

### 3.4. Common Systems Activated by the RISs for Response to PWN Infection

To examine the characteristics of the SAR (induced by the ASM and MeSA treatments) with respect to defence mechanisms, upregulated DEGs were assessed and a comparative transcriptome map consisting of co-expression and unique expression patterns was constructed. The majority (650 genes) of upregulated DEGs were observed to be commonly expressed in response to both ASM (59.6% of the total) and MeSA (57.1%) treatments (Figure 3a). In addition, 441 and 489 genes were found to be uniquely expressed in response to the ASM and MeSA treatments, respectively (Figure 3a). On the basis of BRITE hierarchy analysis, it was found that genes showing co-expression patterns are associated with major biological functions involved in metabolic and signalling processes in pine trees (Appendix A). In contrast, among the DEGs showing a unique expression pattern, the unclassified metabolism category, the molecular relationships of which are unknown [57], accounted for a relatively higher distribution (9.5% and 12.1% following the ASM and MeSA treatments, respectively) than in DEGs showing co-expression patterns. The high proportion and clear function of DEGs showing co-expression patterns under both SAR conditions are indicative of a tendency towards a fundamental defence strategy against PWN infection.

In the further analysis of the two in vivo transcriptomes, DEGs were functionally analysed with reference to the KEGG pathway database, and 161 and 139 pathways in KEGG were assigned to upregulated DEGs in ASM and MeSA treatments, respectively. In line with expectations, DEGs showing co-expression patterns were found to be associated with a wide range of metabolic pathways, among which, 11 pathways were significantly enriched with hypergeometric distribution (FDR value < 0.05) (Figure 3b). Photosynthesis-antenna proteins (ko00196, 6 genes), porphyrin and chlorophyll metabolism (ko00860, 14 genes), glyoxylate and dicarboxylate metabolism (ko00630, 13 genes), carbon fixation in photosynthetic organisms (ko00710, 11 genes), and glycine, serine, and threonine metabolism (ko00260, 12 genes) were the major common pathways activated by ASM and MeSA treatments. In addition, the following secondary metabolite-related pathways were found to be highly enriched: metabolism of xenobiotics by cytochrome P450 (ko00980, 14 genes); drug metabolism-cytochrome P450 (ko00982, 14 genes); stilbenoid, diarylheptanoid, and gingerol biosynthesis (ko00945, 6 genes); flavonoid biosynthesis (ko00941, 11 genes).

Subsequently, the upregulated DEGs showing co-expression patterns were examined to determine significant genetic interactions based on direct and indirect protein–protein associations [60]. In this regard, interaction network maps can make a valuable contribution in gaining a further understanding of the functioning flow of biological systems under conditions of SAR induction. At an interaction cut-off value of 0.5, a total of 561 upregulated DEGs with 409 protein nodes were extracted. After removing non-interacting genes, 175 protein nodes were included in the final network, which consisted of 386 interactions (Figure 4a). Overlaying the functional KEGG information onto the interaction network map revealed a densely connected subnetwork comprising major pathways that are highly enriched in DEGs showing co-expression patterns (Figure 3b). This subnetwork comprised genes associated with photosynthesis-antenna proteins (5 nodes), carbon fixation in photosynthetic organisms (7 nodes), glyoxylate and dicarboxylate metabolism (9 nodes), glycine, serine, and threonine metabolism (6 nodes), and peroxisome (4 nodes). As shown in Figure 4b, glyoxylate and dicarboxylate metabolism in photosynthetic organisms are positioned in the centre of the subnetwork, connected to each of the other three pathways. Among these, PITA_09445 (sedoheptulose-1,7-bisphosphatase) of carbon fixation in photosynthetic organisms showed the highest node degree with 11 interactions. It was also found that PITA_31477, encoding glutamate-glyoxylate aminotransferase 2, and PITA_33653, encoding serine-glyoxylate aminotransferase, occupy critical hub positions associated with the carbon fixation in photosynthetic organisms; glyoxylate and dicarboxylate metabolism; glycine, serine, and threonine metabolism; peroxisome.

Contrastingly, it was found that genes involved in flavonoid biosynthesis constituted an independent subnetwork, comprising eight nodes of 14 upregulated DEGs and 14 interactions (Figure 4c). Without exception, strong evidence was obtained for each gene in the subnetwork, with an average of 3.4 interactions. Moreover, it was found that some reactions within flavonoid biosynthesis are concentrated and reinforced by those genes at a higher resolution (Figure 5). The expression of constituent genes encoding necessary enzymes was found to be highly induced in response to both ASM and MeSA treatments. In particular, PITA_24698 and PITA_38015 (flavonol synthase) genes showed the highest increases in expression, with average changes of 15.4-fold and 22.1-fold, respectively, compared with the control. These genes were found to be commonly associated with the biosynthesis of kaempferol, quercetin, and myricetin. In addition, the reactions yielded different types of flavonoid, some of which are associated with flavone and flavonol biosynthesis (ko00944). The co-expression patterns obtained from the comparative in vivo transcriptomes were confirmed by qPCR analysis. A total of 16 upregulated DEGs were selected from major pathways, including glyoxylate and dicarboxylate metabolism, glycine, serine, and threonine metabolism, peroxisome, and flavonoid biosynthesis. The overall correlation coefficient between RNA-seq data and qPCR data was 0.823, indicating the reliability of the in vivo transcriptome results (Appendix A).

### 3.5. Functional Systems of ASM- and MeSA-Specific Patterns

Results relating to the enrichment of DEGs showing unique expression patterns are presented in Figure 3c. Three and six pathways were found to show significant specificity in response to the ASM and MeSA treatments, respectively. Interestingly, genes associated with flavone and flavonol biosynthesis (ko00944) were commonly identified in both unique groups, despite their different genetic composition. Further investigations of the functional details of genes enriched within flavone and flavonol biosynthesis identified differential selection of flavonoid 3′,5′-hydroxylase (F3′5′H) and flavonoid 3′-monooxygenase (F3′H) as the key enzymes for flavonol biosynthesis. On the basis of reference to previous studies [61,62,63], a schematic diagram was produced, depicting flavonol biosynthetic reactions and the expression levels of related genes (Figure 6). The conversion of naringenin to dihydrokaempferol is the initial step in flavonol biosynthesis and it has been shown that the seven genes of F3′5′H, which catalyses the conversion of dihydrokaempferol to myricetin, were highly expressed in response to the ASM treatment, whereas these genes showed downregulated or non-significant expression in response to the MeSA treatment. In contrast, F3′H (PITA_36657 and PITA_39408) associated with quercetin production were identified as upregulated DEGs only in response to MeSA treatment.

The upregulated DEGs were also classified based on GO terms. Among the GO terms assigned to ASM- and MeSA-specific patterns, the 10 most significant results are listed in Table 2; Table 3, respectively (FDR < 0.05). Responses to various stimuli and substrates were mainly enriched with those DEGs showing ASM-specific expression patterns, including negative regulation of response to stimulus (GO:0048585), defence response to fungus, incompatible interaction (GO:0009817), cellular response to indolebutyric acid stimulus (GO:0071366), and response to abscisic acid (GO:0009737) (Table 2). In addition, there were a large number of DEGs related to plant cell walls, including defence response by callose deposition in cell wall (GO:0052544, 26 genes) and regulation of lignin biosynthetic process (GO:1901141, 17 genes). Similar to the DEGs showing ASM-specific expression patterns, the DEGs upregulated in response to MeSA were associated with a number of response GO terms, including defence response to fungus, incompatible interaction (GO:0009817), response to alcohol (GO:0097305), and response to cold (GO:0009409) (Table 3). In particular, defence response to nematode (GO:0002215) and response to nematode (GO:0009624) terms represented a direct association with parasitic nematodes. Taken together, these data indicated that the unique DEGs induced by the two different RISs tend to be functionally consistent but are associated with different reactions and outputs within a similar system of resistance to PWN in pine trees.

## 4. Discussion

The present study was conceived with the aim of gaining a better understanding of the dynamic interactions between host pine trees and PWN under conditions of SAR induced by treatment with ASM and MeSA. Comparative in vivo transcriptomes allowed access to key traits involved in direct and indirect defence mechanisms of pine trees among many DEGs that predominantly reflect the stressful conditions caused by PWD. Biological phenotypes are typically complex and often involve the control of dozens to hundreds of genes, changes in the expression of which underlie the adaptive processes of phenotypes in response to a range of environmental and developmental factors [64,65]. The study of in vivo transcriptomes by characterising the complete set of transcripts under physiological conditions can be considered to be indicative of a living organism’s blueprint [66]. Given that studies focussing solely on isolated contributary factors often fail to yield meaningful insights into the important elements of biological processes, we designed the present study with a view towards characterising an integrated environment comprising pine hosts, RISs, and PWNs. Simple treatment using RISs is limited to the detection of general genes (termed pathogenesis-related genes) under SAR conditions [67,68], whereas PWN infection in pine trees predominantly reflects the pathogenic mechanisms within dying cells [69]. Consequently, the integrated design of the present study enabled us to assess the sequential stages of RIS treatment and infection, as well as the activation of related genes under conditions of SAR induction, thereby facilitating subsequent analyses of the essential mechanisms underlying the protection against and inhibition of PWN.

As a first line defence response in plants, SAR is an induced immune mechanism that confers a long-lasting state of alert and broad-spectrum non-specific resistance against invaders [27,29,30]. On the basis of an in vivo seedling assay, we detected a marked suppression of PWD, revealing the enhanced resistance of pine trees against PWN conferred by SAR induced by treatments with ASM and MeSA (Figure 1). The RISs used in the present study, ASM and MeSA, are functional analogues of SA. ASM, which was the first synthetic chemical developed as an SAR elicitor, induces a downstream step in SA accumulation [70,71], whereas MeSA, which is synthesised from plants, is a biologically inactive derivative of SA that is converted to SA in distal tissues [72]. These are structurally stable compounds that are not susceptible to chemical modification, such as rapid glycosylation of SA [33,73,74], and are advantageous in that they can readily penetrate and disperse throughout non-infected plants upon exogenous application [73,75]. Thus, applications of ASM and MeSA can effectively mimic the role of SA. Given the origins and roles of these two RISs, the substantial co-expression patterns obtained following exposure to the two RISs under conditions of PWN infection are perhaps predictable (Figure 3). Such co-expression patterns can reveal important functions of conserved regulatory processes across different conditions, environments, or species [76,77]. During PWN infection, these genes may contribute to fundamental strategies required for the protection and survival of pine hosts.

The most significant results obtained with respect to the co-expression patterns indicated the importance of the photosynthesis system (ko00196), carbon fixation (ko00710), and chlorophyll metabolism (ko00860) (Figure 3). Contrastingly, however, numerous studies have revealed that in response to the establishment of SAR, genes involved in photosynthesis are downregulated to confer a fitness advantage in defence signalling under high disease pressure [78,79,80]. At the advanced stage of PWD, nematode-induced embolism causes a decrease in leaf water potential from root tissue, and results in the cessation of photosynthesis [81]. Furthermore, parasitic nematodes have a pronounced effect on the uptake of water and nutrient elements required for photosynthesis [82,83]. Hence, we believe that in RIS-treated pines, the genes associated with the photosynthesis system show a relative upregulation compared with that in dying untreated control pines, rather than reflecting a direct response to SAR induced by the ASM and MeSA treatments. Given the essential role of photosynthesis in plant life, and the high sensitivity of photosynthetic processes to different biotic and abiotic stresses, the effective functioning of photosynthesis, as revealed by in vivo transcriptome analysis, is assumed to be indicative of the establishment of an efficacious defence response against nematode infection.

On the basis of our analyses, we noted several systems, including glyoxylate and dicarboxylate metabolism (ko00630), peroxisome (ko04146), and glycine, serine, and threonine metabolism (ko00260), that interact directly with photosynthesis in co-expression patterns (Figure 4b). Interestingly, most upregulated DEGs distributed in the glyoxylate and dicarboxylate metabolism category are associated with the function of the photorespiration module (M00532) in the entire molecular network (Appendix A). As an energy sink for excess ATP and reducing equivalents, photorespiration represents an important alternative pathway for maintaining the growth of C3 plants such as tobacco, soybeans, and most trees, particularly under stress conditions that lead to reduced rates of photosynthetic reaction [84,85]. It has been reported that the generation of reactive oxygen species (ROS) increases rapidly during photosynthesis under drought/water stress conditions [86,87]. Accordingly, given the negative effects of PWN on water uptake, it is assumed that PWD would induce ROS production in pine trees, causing global damage to cellular components. Photorespiration, peroxisome metabolism, and glycine biosynthesis can provide a functional linkage that contributes to protection against ROS-induced damage. Labudda et al. revealed that the number and size of peroxisomes, which contain enzymes required for photorespiration, are increased in the mesophyll cells of leaves in response to parasitic-nematode infection [88]. Compared with water-sufficient conditions, in plants exposed to drought stress, large amounts of glycine are produced, as a consequence of the photorespiration reactions of the elevated numbers of peroxisomes [89]. Notably, glycine is used as a substrate for the production of glutathione, a key component of antioxidant defence systems [90]. Thus, promoting photorespiration under SAR conditions could serve as an alternative strategy for the survival of pine trees infected with PWD, and the functions of the aforementioned linkage systems could explain the essential interactions related to defence against the cellular damage caused by PWN.

Unlike the immune response in animals, plant immunity lacks a complex adaptive system or specialised cells such as lymphocytes (B and T cells) [91,92]. Plants require less complex immune strategies based on fundamental mechanisms, which are, nevertheless, effective against diseases caused by various invaders or abiotic stresses [93,94]. Accordingly, it would be reasonable for plants to employ bioactive secondary metabolites such as flavonoids, which act as physiological regulators of defence systems, and there have been studies showing the SA dose-dependent production of flavonoids in plants [95,96]. In the present study, we found that under conditions induced by both ASM and MeSA, flavonoid biosynthesis (ko00941) was strongly activated by PWN infection, with the expression levels of associated genes showing an average increase of five-fold (Figure 5). Flavonoids not only function as a secondary antioxidant system by locating and neutralising ROS [97], but also directly and indirectly affect the fitness of nematodes at different life stages [98,99]. In this respect, flavonoids are characterised by nematicidal activity, and Faizi et al. have revealed that treatment of the cyst nematode *Heterodera zeae* with different concentrations of patuletin, patulitrin, and rutin results in high mortality [100]. Furthermore, several flavonoids with similar structural motifs (fisetin, galangin, isorhamnetin, and morin) have been shown to be effective in killing the root-knot nematode *Meloidogyne incognita* at the juvenile stages after a 24-h incubation [101]. Flavonoids can also induce quiescence in nematodes by retarding their motility, which provides a temporal window of opportunity for the mobilisation of defence mechanisms in hosts [98]. In this regard, as repellents of *M. incognita*, quercetin and myricetin have been demonstrated to be highly inhibitive of motility [99]. Furthermore, by limiting egg production or controlling the ratio of males to females, flavonoids can affect the levels of nematode fertility [102,103]. Given the pronounced and wide-ranging effects of flavonoids as nematicidal and nemastatic compounds, our results may provide new evidence highlighting the prominent role played by these compounds in the control of PWN infection in pine trees.

Surprisingly, we found that the fundamental defence strategies deployed by pine trees against PWN infection appear to involve different branches of flavonol biosynthesis downstream of flavonoid production. In the biosynthesis of the myricetin and quercetin, the expression patterns of F3′5′H and F3′H indicated a unique induction in response to only one of the two RISs and showed either non-significant or downregulated expression in response to the other (Figure 6). As representative flavonols, myricetin and quercetin, play an important role in resistance against parasitic nematodes by functioning as nematicidal and nemastatic compounds [99]; however, details of the functional differences between these two flavanols are yet to be ascertained. It has been reported that despite the same SA treatment, wheat plants show different amounts and fluctuations in the production of myricetin and quercetin, depending on the treatment method or plant tissues examined. These observations imply that myricetin and quercetin have independent functions under different conditions. Thus, we hypothesise that the production of different flavonols in response to ASM and MeSA treatment constitutes an important facet of the defensive mechanisms against PWN infection. Furthermore, in the GO analysis of the ASM-specific patterns, we identified the defence response (GO:0052544) and regulation of lignin biosynthesis (GO:1901141) in the plant cell wall as processes that were highly enriched (Table 2). We established that the defence response to nematode (GO:0002215) was the most significantly enriched process associated with the MeSA-specific expression patterns, among which seven genes encode glucan endo-1,3-β-glucosidases (Appendix A), which, as important components of plant cell walls, play roles in defence against pathogens [104,105]. Other genes encoding peroxidases (Appendix A) are also involved in the defence mechanisms of plants via cell wall reinforcement [106]. Given that cell wall structure has been implicated as an effective physical barrier against invaders, ASM and MeSA treatments are expected to induce different defence responses based on plant cell wall modification.

## 5. Conclusions

In conclusion, in this study, we performed comparative in vivo transcriptomics in order to characterise the fundamental defence strategies deployed by pine trees against PWN infection. We found that application of the two SAR elicitors, ASM and MeSA, was effective in terms of reducing PWD severity, and induced similar patterns of gene expression associated with plant functions. By overlaying enriched biological pathways onto an interaction network, we demonstrated that the genes showing co-expression patterns constitute a functionally linked system for protection from ROS damage caused by PWN infection: peroxisome → photorespiration → glycine → antioxidant glutathione. Furthermore, we observed that flavonoid biosynthesis was highly activated in response to treatment with the two RISs and could act as a countermeasure strategy induced by pine trees to control PWN. Among the genes showing unique patterns of expression in response to SAR, we identified F3′5′H and F3′H, which are associated with the production of the flavanols myricetin and quercetin, which are assumed to play important roles in defence mechanisms. To the best of our knowledge, this is the first study to interpret the dynamic interactions between host pine trees and PWN under conditions of SAR induced by exogenous RISs. Our study, underlying the efficacies of ASM and MeSA, could suggest the possibility of effective PWN control agents based on the reinforcement of the development of, and resistance in, pine trees.

## Figures and Tables

**Figure 1 genes-11-01000-f001:**
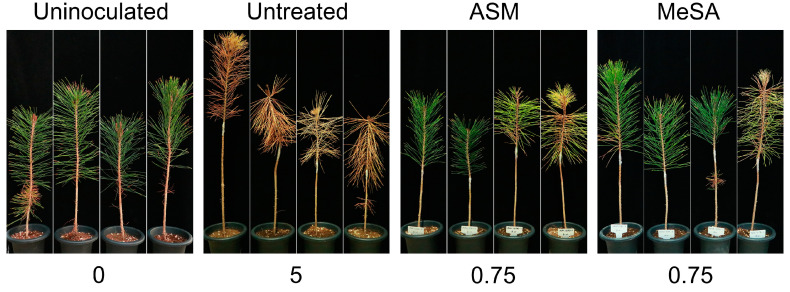
The influence of acibenzolar-*S*-methyl (ASM) and methyl salicylic acid (MeSA) on pine wilt disease (PWD) resistance. Photographs of pine seedling inoculated with the nematode *B. xylophilus* were taken at 30 days after resistance-inducing substance treatment. PWD severity was scored for needle discolouration on a scale of 0 to 5. The numbers below the photographs denote the mean severity scores determined from four seedlings per treatment. Uninoculated seedlings served as positive controls, and untreated seedlings, as negative controls.

**Figure 2 genes-11-01000-f002:**
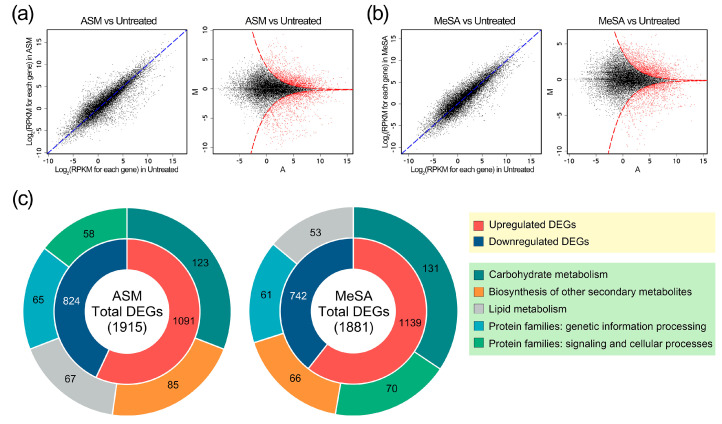
Overview of the differentially expressed genes (DEGs) in acibenzolar-*S*-methyl (ASM) and methyl salicylic acid (MeSA) treatments. Differential expression was analysed using scatter plots and M (log ratio) against A (mean average) scales (MA-plots) in the DEGseq package. Each point represents one of 45,528 non-redundant genes in pine trees. (**a**) Comparison of the ASM-treated and untreated samples. The scatter plot depicts differences in gene expression between the two groups after normalisation. Genes showing significant differences in expression are indicated by red colour in the MA-plot (false discovery rate, FDR < 0.05). (**b**) Comparison of the MeSA-treated and untreated samples. (**c**) Donut charts illustrating the distribution of DEGs and functional hierarchies. The inner layer shows the distribution of upregulated and downregulated DEGs, and the outer layer comprises the top five functional categories. The size of ring segments is indicative of the relative proportion of DEGs in each category.

**Figure 3 genes-11-01000-f003:**
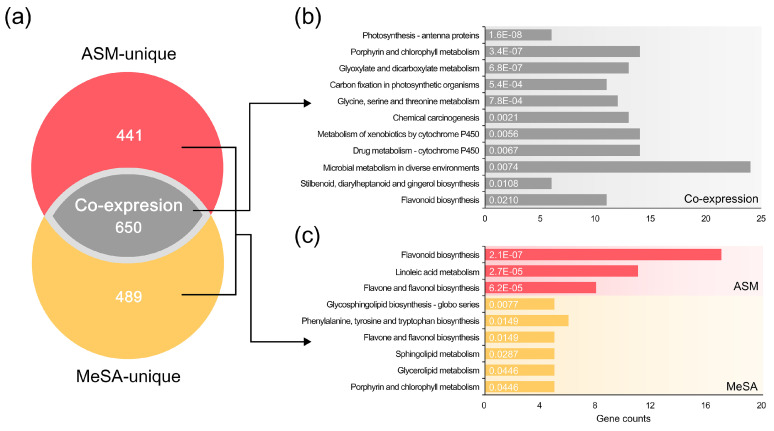
Comparative in vivo transcriptomes of upregulated differentially expressed genes (DEGs) and Kyoto Encyclopedia of Genes and Genomes (KEGG) pathway enrichment in acibenzolar-*S*-methyl (ASM) and methyl salicylic acid (MeSA) treatment groups. (**a**) A Venn diagram showing the co-expression and unique expression groups of upregulated DEGs. Comparative analysis revealed 650 overlapping DEGs. A further 441 and 489 DEGs were identified as being uniquely expressed in response to the ASM and MeSA treatments, respectively. (**b**) Significantly enriched KEGG pathways for the co-expression group. Significant pathways were identified and ranked based on hypergeometric distribution with an FDR < 0.05. The length of bars indicates the number of DEGs in each KEGG pathway. (**c**) Significantly enriched KEGG pathways for ASM- and MeSA-specific patterns.

**Figure 4 genes-11-01000-f004:**
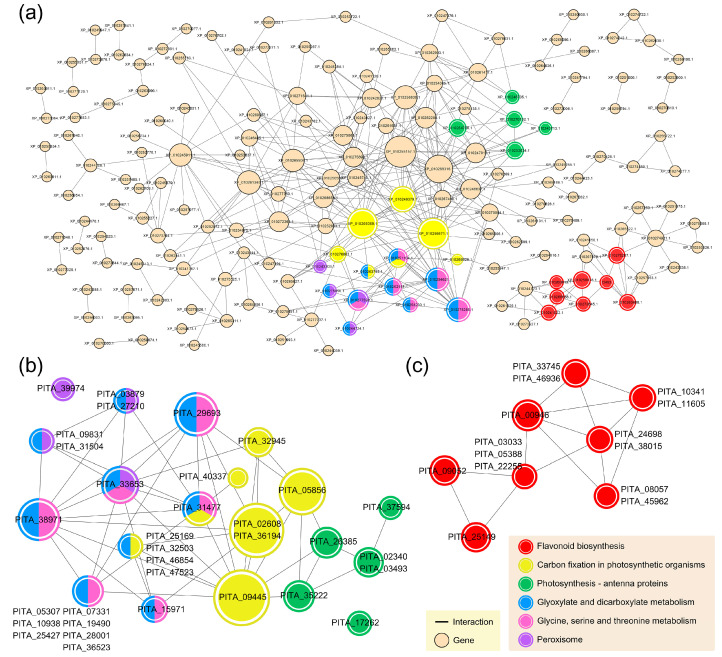
A network of co-expression patterns based on protein–protein interactions. (**a**) The overall network structure with pathway information. For the co-expression patterns, protein–protein interaction information was collected using Search Tool for the Retrieval of Interacting Genes/Proteins (STRING) v11 and visualised using the Cytoscape tool. A total of 650 upregulated DEGs were used, with interactions showing a combined score of less than 0.5 being filtered out, resulting in 175 nodes and 386 edges. The diameter of the node reflects the number of direct connections to other nodes. Within nodes, KEGG pathways are indicated by different colours (identified in the accompanying key). If genes are associated with more than two KEGG categories, the nodes are divided into the corresponding colours. Two groups of densely connected pathways are displayed as subnetworks. (**b**) Carbon fixation in photosynthetic organisms; photosynthesis-antenna proteins; glyoxylate and dicarboxylate metabolism; glycine, serine, and threonine metabolism; peroxisome. (**c**) Flavonoid biosynthesis.

**Figure 5 genes-11-01000-f005:**
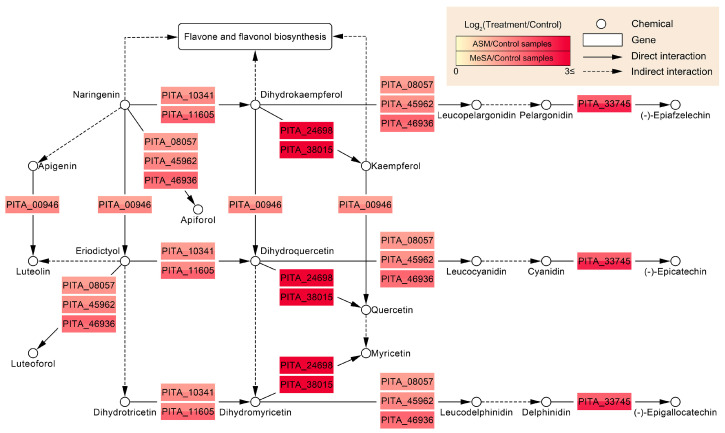
Molecular networks with expression changes in flavonoid biosynthesis. The nodes illustrate the type of molecules as follows: genes (coloured rectangles); biochemicals (circles); cognate pathway (white rectangle). The differences in colour are indicative of the corresponding changes in log_2_(treatment/control) values from a minimum of 0 to a maximum of 3, where a value of 0 indicates the same expression level between treatment and control conditions. The direction of biosynthesis in a pathway is represented by arrows adjacent to the gene names.

**Figure 6 genes-11-01000-f006:**
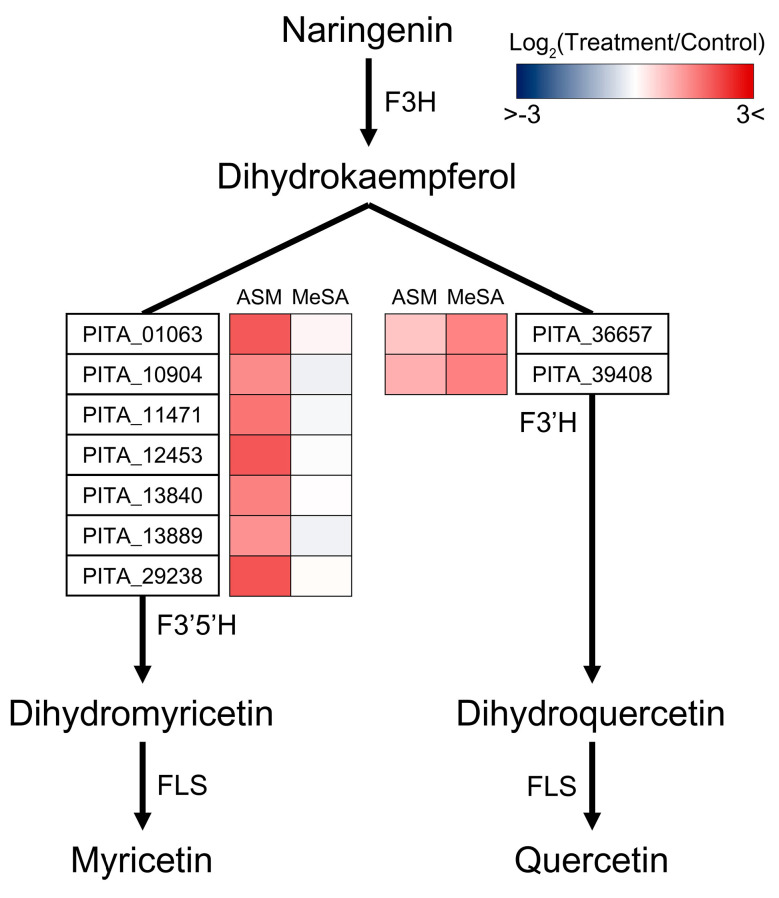
Schematic diagram of flavonol biosynthesis-related genes showing unique expression patterns. The biosynthetic diagram for the production of myricetin and quercetin from naringenin has been modelled. In the acibenzolar-*S*-methyl (ASM)- and methyl salicylic acid (MeSA)-specific patterns, the expression levels of genes have been calculated according to the log_2_(treatment/control) values and are shown as heatmaps. Each coloured cell marked in red indicates upregulation and each cell in blue indicates downregulation. Abbreviations: F3H, naringenin 3-dioxygenase; F3′5′H, flavonoid 3′,5′-hydroxylase; F3′H, flavonoid 3′-monooxygenase; FLS, flavonol synthase.

**Table 1 genes-11-01000-t001:** Detailed information of the RNA-seq libraries.

Sample	Read Type	GC (%)	Quality ^1^	Total Reads	Total Length (bp)
ASM_1	Paired end	47	40	27,814,772	2,805,933,290
ASM_2	Paired end	45	40	35,228,444	3,554,276,515
ASM_3	Paired end	46	40	33,425,670	3,372,510,853
MeSA_1	Paired end	45	40	31,973,654	3,226,375,119
MeSA_2	Paired end	45	40	33,377,584	3,367,938,501
MeSA_3	Paired end	46	40	33,589,898	3,389,002,001
Untreated_1	Paired end	45	40	31,763,374	3,205,080,196
Untreated_2	Paired end	46	40	33,120,212	3,341,300,502
Untreated_3	Paired end	45	40	34,069,162	3,437,840,259

^1^ Average Phred quality score.

**Table 2 genes-11-01000-t002:** Gene ontology (GO) enrichment of acibenzolar-*S*-methyl-specific patterns.

GO ID	Terms	Count	FDR ^1^
GO:0048585	Negative regulation of response to stimulus	34	0
GO:0006855	Drug transmembrane transport	26	0
GO:0009817	Defence response to fungus, incompatible interaction	26	0
GO:0042344	Indole glucosinolate catabolic process	26	0
GO:0052544	Defence response by callose deposition in cell wall	26	0
GO:0070574	Cadmium ion transmembrane transport	26	0
GO:0071366	Cellular response to indolebutyric acid stimulus	26	0
GO:1901140	*p*-Coumaryl alcohol transport	17	0
GO:1901141	Regulation of lignin biosynthetic process	17	0
GO:0009737	Response to abscisic acid	44	6.44 × 10^−11^

^1^ False discovery rate.

**Table 3 genes-11-01000-t003:** Gene ontology (GO) enrichment of methyl salicylic acid-specific patterns.

GO ID	Terms	Count	FDR ^1^
GO:0002215	Defence response to nematode	9	1.27 × 10^−7^
GO:0009624	Response to nematode	14	1.16× 10^−5^
GO:0006833	Water transport	10	2.37 × 10^−5^
GO:0009094	l-Phenylalanine biosynthetic process	5	2.37 × 10^−5^
GO:0003333	Amino acid transmembrane transport	7	0.0056
GO:0009817	Defence response to fungus, incompatible interaction	8	0.0096
GO:0015804	Neutral amino acid transport	5	0.0096
GO:0051704	Multi-organism process	40	0.0407
GO:0097305	Response to alcohol	25	0.0443
GO:0009409	Response to cold	14	0.0443

^1^ False discovery rate.

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
