# Peer review of "Comparative Transcriptome Analysis of Pine Trees Treated with Resistance-Inducing Substances against the Nematode Bursaphelenchus xylophilus"

_genes, 2020, doi:10.3390/genes11091000_

Round 1
Reviewer 1 Report
The authors report on the comparison of the transcriptome induced by the treatment of pine infected with the pinewood nematode, with two SAR stimulating compounds. The stimulation of plant natural defense systems against nematode diseases is a promising sustainable strategy to improve current pest management practices. I congratulate the authors on both the written form and the extensive work performed for this publication. The overall manuscript is presented in a straightforward scientific manner, the methodology is presented simply and directly. It is a valuable work towards advancing the current knowledge on resistance of pine to the pine wilt disease.
Nevertheless, the manuscript shows some problems that must be addressed before it is acceptable for publication.
The authors must be extremely careful in extracting conclusions towards specific mechanisms used against the PWD since treatment results were only compared with the transcriptome of heavily affected trees (pines with strong pinewood nematode infection symptoms, called untreated) and differences in DEGs could just be reflective of a healthier plant. In a highly diseased state, the transcriptome reflects a strongly stressful state and the mechanisms therein are very specific. You should stress this fact throughout the text so the reader can be aware of this.
Throughout the document (using the document line numbering)
Line 121 – Please explain what constitutes CR-KSP40M and CR-MOC25, and why you used the several formulation components.
Line 233 – Perhaps the explanation of the whole scale used to characterize PWD severity is too long. I suggest describing the lowest and the highest value.
Line 301 – Consider deleting “These observations revealing,”.
The present manuscript presents extensive and important results and should be considered for publication.
Reviewer 2 Report
Pine wood nematode (PWN), Bursaphelenchus xylophilus causes Pine wilt disease (PWD). The use of SARA would overcome the use of pesticides. Detailed details pathways that underlie PWN resistance remain uncertain. In this manuscript, the authors examined impact of two inhibitors of the diseases symptoms, ASM and MeSA, upon Bursaphelenchus xylophilus infection. Three-year old seedlings of Pinus densiflora were treated with ASM and MeSA and after 1 week with PWN. Degree of needle discoloration was sign of severity of the diseases. After 30 days of experiment, a positive effect of Pinus treatment with ASM and MeSA, inhibition of the diseases symptoms was shown. Performed in vivo transcriptome analyses related patterns of the functional gene expression upon ASM and MeSA treatment was revealed.
The article is well structured, written with clear and concise scientific language. The title describes the contents of the article. The all parts of article (The abstract, the material and methods, the results, the conclusion and the references) are adequately described. The article provided new knowledge about the pines response to infection with pine wood nematode after treatment with SAR elicitors.
I recommend the article for publication.
I suggest the following corrections:
Line 226: I suggest to state infection rates of pine as a percentage with a score of O.75.
Line 260: choose word writing: cutoff or cut-off (Line 332).
